# MapQA: A Map-Question-Answering Benchmark for Visual Language Model Reasoning

## Abstract

Maps are central to how humans make sense of the world, from navigation and environmental monitoring to military planning and historical interpretation. Yet despite rapid progress in large multimodal models (LMMs), these systems continue to struggle with interpreting maps–an essential skill for visual reasoning that goes beyond pattern recognition and text extraction. To close this gap, we introduce MapQA, the first large-scale benchmark specifically designed to evaluate LMMs on map understanding. MapQA contains over 4,200 carefully curated, open-ended question–answer pairs spanning diverse map types, each constructed to require reasoning directly from the map rather than relying on memorized world knowledge. Benchmark questions are generated through a scalable human-in-the-loop process to ensure quality, and evaluated using an LLM-as-a-judge protocol aligned with human judgments. Our experiments show that while humans answer over 91% of questions correctly, state-of-the-art proprietary models achieve barely half that performance, with open-source models typically below 30%. These findings highlight a substantial gap between human and machine map understanding, underscoring the need for benchmarks like MapQA to guide future progress in multimodal reasoning.

## 1 Introduction

LMMs have shown remarkable improvement in open-domain visual question answering across a broad range of tasks, such as math, photography, and interpreting charts and diagrams (Agrawal et al., 2024; Abdin et al., 2024; OpenAI, 2024a). A wide variety of benchmark datasets, spanning from general perception (Yue et al., 2024; Ying et al., 2024) to specialized tasks (Mathew et al., 2021; Lu et al., 2023; Masry et al., 2022), exist to evaluate LMMs' capabilities. However, to achieve strong perception across domains, LMMs must extend into more specialized and challenging areas that often lack comprehensive benchmark datasets.

Map understanding is one such domain of particular importance. This is because maps aid our understanding of the world and shape our perception through the narratives they construct (Marthew, 2023). They have broad applications throughout society, including ecological and geoscientific maps for understanding Earth systems (Teixeira & Teodoro, 2021), tactical cartography for military situational awareness (Liebenberg et al., 2016), and street maps for navigation (Nakhimovsky et al., 2010). Maps are widely used because they act as external cognitive aids, offloading abstract and complex concepts, as well as spatial relationships, into symbolic representations that humans can quickly interpret (Casati, 2023).

Despite recent advances, LMMs continue to struggle with understanding and interpreting maps. Tong et al. (2024) alludes to the reasons for this difficulty by identifying nine categories of common failure for LMMs in answering questions: (i) orientation and direction, (ii) presence of specific features, (iii) state and condition, (iv) quantity and count, (v) positional and relational context, (vi) color and appearance, (vii) structural and physical characteristics, (viii) text, and (ix) viewpoint and perspective. All of these categories are foundational to map understanding, as they capture the essential information and representational structures through which maps convey meaning and support interpretation. In addition, Goyal et al. (2017); Rohrbach et al. (2018) show that LMMs often fail to fully incorporate the image in their processing. Instead, they rely on immediate language input and the intrinsic knowledge learned from language during training. With the complex and abstract

relationships encoded in maps, reliance on language and historical human knowledge can lead to inaccurate responses or hallucinations. Figure 1 shows an example of GPT-5 incorrectly relying on historical information from language rather than fully incorporating the map signal.

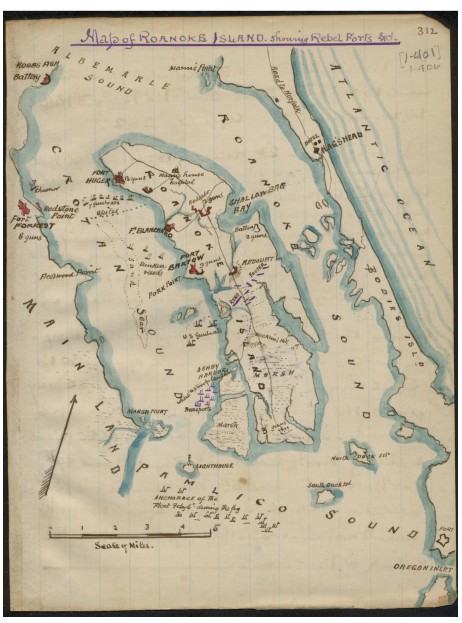

**Context:** The Confederates (red) abandoned Fort Huger to Union forces (purple) during the Battle of Roanoke Island (February 1862). However, the map shows the positions before the retreat, and thus the historical outcome of the battle had not yet occurred on the map.

**Question:** Are the Confederates abandoning Fort Huger during the Battle of Roanoke Island?

**Correct Answer:** No, the Confederates have not abandoned Fort Huger.

**GPT-5 Answer with the map:** Yes, during the Battle of Roanoke Island on February 8, 1862, the Confederates abandoned Fort Huger after Union forces overran their defensive positions.

**GPT-5 Answer without the map:** Yes, during the Battle of Roanoke Island (February 1862), the Confederates abandoned Fort Huger after Union forces broke through their defenses and cut off their retreat.

Figure 1: Battle of Roanoke Island: Image with context, question, ground truth, and GPT-5 incorrect answers. The inclusion of the date and the outcome of the battle in both answers indicates that outside contextual information was used, and the map was not fully analyzed or was disregarded.

To address these issues, we present the MapQA Benchmark, a new vision-based question answering dataset designed to evaluate LMMs' reasoning skills on a diverse set of complex, information-rich maps. The questions in the dataset were specifically designed and curated to be answerable only with the map, in order to directly test map understanding and prevent LMMs from taking shortcuts using intrinsic knowledge from language. The dataset was created through human-in-the-loop LLM generation and careful curation of generated outputs and human-annotated answers. We evaluate the benchmark on a wide variety of state-of-the-art open-source and proprietary models and demonstrate their limitations in map question answering.

## 2 RELATED WORK

**Map Benchmark Datasets** With the growth of research into LMMs, single-task (Antol et al., 2015; Goyal et al., 2017; Marino et al., 2019; Lin et al., 2014) and general-perception (Yin et al., 2023; Yu et al., 2023; Liu et al., 2024b; Yue et al., 2024) multimodal datasets have become increasingly important for training and evaluation. While the latter provide more holistic coverage, single-task evaluations remain critical for probing isolated skills of LMMs. This is because domain-specific sub-capabilities may be overlooked if certain question types are not framed appropriately or are omitted entirely (Li et al., 2025). Within this space, map question-answering remains limited in scope. Extensive map datasets exist across domains such as geography (U.S. Geological Survey, 2025b; Kelso & Patterson, 2009; Fick & Hijmans, 2017), remote sensing (U.S. Geological Survey, 2025a; Hansen et al., 2013; Van Etten et al., 2018), transportation (OpenStreetMap contributors, 2017; Chang et al., 2019), and history (Library of Congress, 2025; David Rumsey Map Collection, 2025; Kramm et al., 2025). However, these are largely uncurated collections of raw data without labels or paired QA annotations needed to evaluate LMMs. Prior work has examined visual question answering for maps, but has typically restricted its focus to remote sensing, geometric questions, choropleth maps, and transportation maps rather than broader applications (Lobry et al., 2020; Chen et al., 2021a; Chang et al., 2022; Li et al., 2025; Punjani et al., 2018; Kefalidis et al., 2023) .

**Algorithmic QA-Judge** Building algorithms that can effectively judge answers is an active area of research. Two common techniques are key-phrase matching and using LLMs as judges. Key-phrase matching aligns key phrases, words, or tokens in the ground-truth answer with those in the generated response. Early work primarily relied on regular expressions (Yue et al., 2024; Sychev, 2023), while later approaches assign token importance through weighting (Lee et al., 2021) or neural classifiers (Bulian et al., 2022). These methods still face challenges such as varying answer granularity, multiple candidate answers, and lexical mismatches. Some approaches attempt to address these issues by expanding ground-truth knowledge sets (Si et al., 2021) or by applying natural language inference models to reformulated questions and answers (Chen et al., 2021b; Honovich et al., 2021). However, they remain limited when key phrases fall outside the dataset.

LLMs as judges have seen growing interest in recent years for their potential to overcome the flaws of key-phrase matching. Popular approaches involve directly prompting an LLM to evaluate QA responses (Kamalloo et al., 2023; Myrzakhan et al., 2024). For example, Myrzakhan et al. (2024) propose prompts for evaluating both open-set and multiple-choice answers, which we adopt for our MapQA evaluation task. Other work has examined LLM-based evaluation of natural language generation and summarization (Liu et al., 2023; Sottana et al., 2023), while Zheng et al. (2023) explore using an LLM to rank chatbot answers by correctness and quality. Since large models are expensive to run, recent efforts have sought to reduce costs by finetuning smaller LLMs on judgments from larger models such as GPT-4o (Zhu et al., 2023).

## 3 THE MAPQA BENCHMARK

### 3.1 OVERVIEW

We introduce the MapQA Benchmark, a novel vision-based question-answer benchmark to test LMMs ability to reason over information-dense images, and evaluate current model capabilities. Human perception relies on vision at its core, and function as a primary method of communication. Where text can be laborous to read and interpret, a map is easy to understand and presents massive quantities of information to a user in a manner that is quickly interpretable while also not being overwhelming. To further improve models and thoroughly test their capabilities, a rigorous vision-based dataset is needed to compare models against human commonsense and reasoning. The MapQA dataset encompasses four primary map types—Military, Natural World, Urban, and Aviation—spanning 24 subcategories in total. Each type poses distinct challenges for visual understanding, making them valuable for systematic evaluation.

MapQA is composed of 4,234 questions, with each one designed to be unanswerable without the aid of a map. This ensures we eliminate bias from parameterized knowledge about any given region. For example, questions like "Which airport is closest to Chicago O'Hare?" are easily answerable without a map because the knowledge is very likely to already be in a given model's training data. We also minimize the number of questions that can be answered strictly via Optical Character Recognition (OCR), as these questions are simplying reading text on a given map rather than reasoning over the map and it's implications. Lastly, all questions must be answerable by a reasonable percentage of humans. In our case, we found that among questions that were tested, humans answer 91.4% of them correctly.

The MapQA dataset is an open-answer dataset which tests model's reasoning capability rather than guessing which multiple choice option appears most correct. However, open-answer dataset come with their own challenges, as it is difficult to systematically evaluate when a correct answer is given. For this reason, we use LLM-As-A-Judge. When compared to other models, we found GPT4o to be the most accurate and cost effective. After calibrating GPT4o to over two thousand human judgements of MapQA questions, we fit GPT4o judgements to a linear model $y = \alpha + \beta x$ and found $\alpha = 0.035$ and $\beta = 0.921$, indicating that GPT4o when used as LLM-As-A-Judge, while not perfect, is a very strong indicator of ground-truth accuracy.

To score model predictions we adapt the LLM-As-A-Judge framework modifying the system so that it issues a binary correct/incorrect verdict, tolerating minor phrasing differences as well as numerical slack ($\pm 10\%$) when judging measurement heavy questions. These instructions diverge from the original paper's open form rationales by tightly constraining the response and explicitly allowing

| Statistics | | | | Number |
|---|---|---|---|---|
| Total Questions | | | | 4,234 |
| Total Map Types/Subcategories | | | | 4/24 |
| Splits | Hand-Annotated 1,647 | | Validated 480 | Human-in-the-Loop 2,587 |
| Difficulties | Very Easy 20.7% | Easy 30.3% | Medium 19.3% | Hard 14.2% Very Hard 15.5% |
| Open-Answer Questions | | | | 4,234 (100.0%) |
| Questions with Multiple Images | | | | 811 (19.2%) |
| Average question word length | | | | 12.7 |
| Average answer word length | | | | 4.1 |
| Total Unique Images | | | | 794 |
| Largest image size (pixels) | | | | 6000 x 6000 |
| Smallest image size (pixels) | | | | 546 x 393 |
| Average image size (pixels) | | | | 4782 x 4198 |
| Standard deviation (pixels) | | | | 1545 x 1289 |

Table 1: Key statistics of the MapQA benchmark.

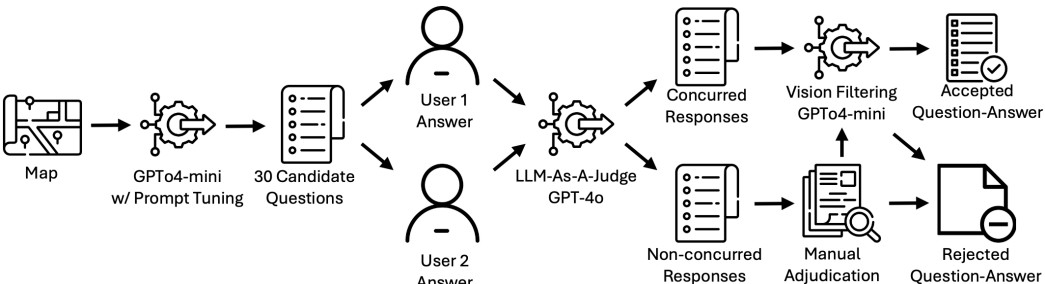

Figure 2: Automated Human-in-the-loop Curation Flow Diagram.

the judge to reconcile student and reference answers that describe the same cartographic features in different words.

## 3.2 CURATION PROCESS

**Hand-Curation** The MapQA benchmark is carefully curated with the previously stated inclusion criteria in mind. For the first 2,000 questions, our co-authors hand-curated every questions and cross-scrutinized each for clarity and interpretability before culling any question that is answerable without a map. Culling was repeated until accuracy fell below 3.5%–the previously computed $\alpha$ for GPT4o when used in LLM-As-A-Judge. After scrutinization and culling, 1.6k questions remained over 172 unique images of maps.

**Human-in-the-loop Curation** Manual curation is inherently labor-intensive and time-consuming, and it can be difficult to monitor quality over a large dataset when annotators tend to converge towards questions that follow repetitive and formulaic patterns. While crowdsourcing offers a potential alternative, our attempts revealed significant limitations: crowdworkers lacked familiarity with the capabilities and shortcomings of modern LMMs, and frequently generated questions that were either answerable without the given map or depended primarily on optical character recognition (OCR). Consequently, fewer than 5% of crowdworker-generated questions satisfied our inclusion criteria, underscoring the need for an alternative strategy.

Instead, we implemented an automated human-in-the-loop curation strategy that leverages few-shot examples for each map type. Starting with an image of a map, we use GPTo4-mini to generate 30 candidate questions based on 50 example questions from our hand-curated set. Those questions are

```
Images presented are tiles from one single image.  The
tiles are presented from top to bottom, left to right.
Tiles are padded with black pixels.  Answer the
question considering all image tiles.  Your answers
should be relatively concise.
```

Figure 3: MapQA System Prompt to answer questions

presented to two annotators who have previously been evaluated for their expertise on the hand-curated set. LLM-As-As-Judge then evaluates whether the answers provided by each annotator is similar. If the answers are similar, then the question-answer pairs are culled until LLM-As-Judge Accuracy without images falls below 3.5%. Once this is complete, the questions-answer-map triplets are accepted into the benchmark set. If the two responses are not found to be similar, they advance to a manual review from our research team where, after manual review and possible revision, the question-answer may be ultimately accepted or rejected. The details of this process are shown in fig. 2. Using this automated curation strategy, we added an additional 2.6k questions for a total of 4.2k questions.

**Finding Expert Humans** We determine an individual annotator's aptitude by first surveying their response to a random subset of our original hand-curated set from the research team. For each map type, we randomly sample approximately 120 questions to determine the calibrated accuracy, $p$, for that annotator. Those with a $p > 0.95$ are therefore considered 'experts' in that category.

For two experts, each with accuracy $p$,     $P(\text{correct} \mid 2 \text{ unanimous}) = \frac{p^2}{p^2+(1-p)^2}$.

$$P(\text{correct} \mid p = 0.90) = \frac{0.90^2}{0.90^2 + 0.10^2} \approx \frac{0.81}{0.81 + 0.01} \approx 0.9878,$$

Thus, if responses are concurred by our annotators that average a 91.4% accuracy on the 120 question set, we have strong statitical confidence that the concurred answer to automatically generated questions are, in fact, correct.

**Determining Difficulty** During the evaluation of annotators on our hand-curated set, we also measure the length of time required to respond to each question. This data is sorted into quintiles from which we derive five difficulty categories: Very Easy, Easy, Medium, Hard, and Very Hard. This relationship is then used to determine the difficulty of all subsequent questions created via the human-in-the-loop data curation process.

## 4 EXPERIMENTS

We consider various large multimodal models to evaluate the MapQA benchmark, with a near even split between open-source models and proprietary models. We choose the latest model version for each family for our evaluation, and use a variety of model parameter sizes within each family to observe how performance scales with model size. We perform zero-shot evaluation with a system prompt to evaluate the model capability to generate correct answers to the benchmark questions without finetuning. The system prompt is show in fig. 3. For open-source models, experiments are conducted with two different types of GPUs depending on model size. For very large models, the model is sharded across 4 NVIDIA H100 80GB GPUs. When a smaller model is possible, the model is either run on a single NVIDIA A100 40GB GPU or it may be sharded across 4 A100 40GB GPUs.

### 4.1 BASELINES

**Open-Source LMMs** We evaluate various open-source models of different parameter sizes. Qwen2.5-VL (i. 2B, ii. 7B) extends the Qwen-VL series by introducing dynamic resolution processing to handle images of various sizes, enabling perception of spatial scales without traditional normalization techniques (Bai et al., 2025). Pixtral-12B has a custom-trained vision encoder that processes images of varying resolutions and aspect ratios (Agrawal et al., 2024). (iii.) Phi-3.5 Vision Instruct ONNX interweaves vision tokens from CLIP ViT-L/14 and text tokens before passing

them into the phi-3.5-mini decoder (Abdin et al., 2024). LLaVA-Next passes visual encodings into an LLM using both fine-grained patch embeddings and a global embedding from the resized image. We evaluate LLaVA-Next with various LLM decoders and sizes: Qwen (iv. LLaVA-Next-34B, v. LLaVA-Next-72B), Vicuna (vi. LLaVA-Next-Vicuna-7B, vii. LLaVA-Next-Vicuna-13B), and Mistral (viii. LLaVA-Next-Mistral-7B) (Liu et al., 2024a). (ix.) Llama-3-11B-VL adds image input support by adding cross-attention adapter weights to a pretrained image model before passing them into a pretrained language model (AI, 2024).

The large images found in MapQA require orders of magnitude more token input than regular text. As a result, some models that can run on the listed devices with text input alone are unable to fit a single sample in the remaining memory of an 80 GB GPU after loading just one-quarter of the model. For this reason, we are unable to benchmark models such as Pixtral-124B, LLaVA-Next-110B, Llama-3-90B, Qwen2.5-VL-32B, and Qwen2.5-VL-72B.

**Proprietary LMMs** We also consider several proprietary models. (x.) GPT-4o is OpenAI's first omni model, capable of accepting and generating any combination of text, audio, image, and video input (OpenAI, 2024a). (xi.) GPT-4.1 improves on GPT-4o with a longer context window and improved instruction following (OpenAI, 2025a). (xii.) GPT-4o-mini is a compact and faster version of GPT-4o (OpenAI, 2024b). (xiii.) GPT-5 is a unified system with a deep reasoning model for harder problems and a router to select the appropriate model given the conversation and prompt (OpenAI, 2025b). (xiv.) GPT-5-mini and (xv.) GPT-5-nano are smaller, more compact versions of GPT-5 (OpenAI, 2025b;c). Anthropic's Claude-Opus-4 is a strong coding model focused on complex reasoning and long-running tasks, while Claude-Sonnet-4 is tuned for improved instruction following in coding and reasoning tasks (Anthropic, 2025). Claude-3.5-Haiku provides lower latency for inference while offering comparable performance to Claude-Opus-3 (Anthropic, 2024).

We attempted to run inference using Gemini, but after processing 1k questions we encountered a file storage quota issue with our Google Cloud account, despite not using any file storage. We have an open ticket with support and hope to complete the evaluation of Gemini once it is resolved.

**Humans** To assess annotation quality, we evaluated all human annotators on a random sample of 120 hand-curated questions from each map type, yielding a validation set of 600 questions in total. The resulting accuracies, disaggregated by category, are reported in table 2.

**Evaluation** We report micro-averaged accuracy, treating each question as a single instance and averaging correctness over all items in the benchmark. Because MapQA is open-answer, we assess correctness using an LLM-as-a-Judge protocol (GPT-4o) with the model's intermediate reasoning masked to prevent leakage. Judge outputs are calibrated to human judgments via a linear mapping $y = \alpha + \beta x$ fit on more than two thousand human-rated responses for each map type; the resulting coefficients are used to inverse-map raw judge scores to calibrated accuracies. Reported accuracies reflect this calibrated, micro-averaged measure over the full evaluation set.

## 4.2 BENCHMARK RESULTS

The MapQA benchmark is tested against 18 models and plotted in fig. 4. On aggregate, humans strongly outperform all models with 91.4% accuracy based on performance on the Validated set of questions. The best models against the benchmark are the proprietary models. GPTo4-mini tops the benchmark with 52.5% accuracy. The best performing open-source model is Qwen2-2B with 26.1% outperforming Claude Sonnet 4 and Haiku 3.5, while slightly underperforming Opus 4. There appears to be little difference between OpenAI generations when comparing GPTo4-mini to

| Category | Human Accuracy |
|---|---|
| Military | 0.937 |
| Natural World | 0.904 |
| Urban | 0.908 |
| Aviation | 0.909 |
| **Benchmark Average** | **0.914** |

Table 2: Average human accuracy across MapQA categories.

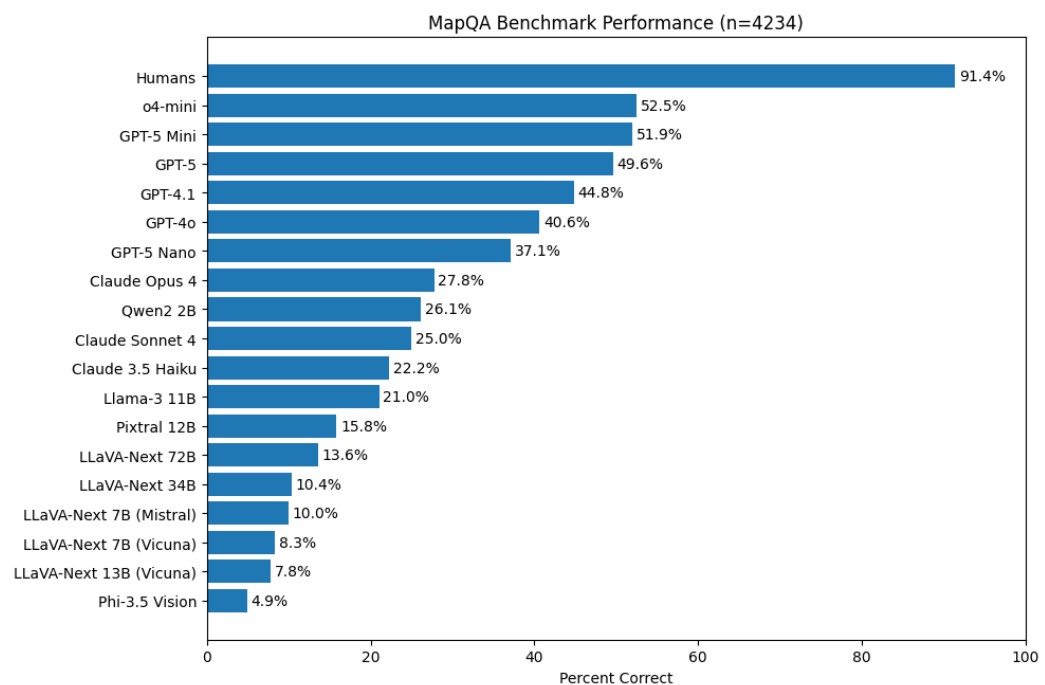

Figure 4: MapQA Benchmark evaluation across all models. Humans significantly outperform the best LMM models.

GPT5-mini, with GPTo4-mini slightly outperforming the latter by 0.6%, and a small improvement from GPT4.1 to GPT5, with a difference of 4.8% separating the two.

Among the Claude family of models, there also does not appear to be large differences between model performance. Based on model pricing, one would assume that Claude Opus 4 would significantly outperform Sonnet 4 and Haiku 3.5. According to the pricing at the time of writing, Opus 4 is 5x more expensive than Sonnet 4, and 18.75x more expensive than Haiku 3.5, however, the difference between Opus 4 and Haiku 3.5 is a meager 5.6%.

## 5 ANALYSIS

We analyze the benchmark to provide insights that could explain model performance and how they can be modified to further improve accuracy. We explore model size, tokens used, and performance group by difficulty.

### 5.1 MODEL SIZE INSIGNIFICANCE

We analyze how MapQA performance scales with parameter size of each model. Unfortunately, there is little information available regarding the size of proprietary models, so only comparisons of open-source model accuracy is performed. This does mean that the comparison is limited because no open-source model exceeds 30% accuracy, however, as shown in fig. 7, there is no statistically significant trend. Qwen2.5-VL 2B is the best performing open-source model, but also the smallest, while LlaVA-Next 72B is the largest and shows middling performance.

### 5.2 POSITIVE CORELATION WITH TOKEN USAGE

When comparing the average number of tokens used against a model's average accuracy, an interesting trend emerges: As the number of average tokens increases, the accuracy against MapQA also increases with a strong (r=0.6976) correlation that is statistically significant across 4.2k samples, as shown in fig. 6.

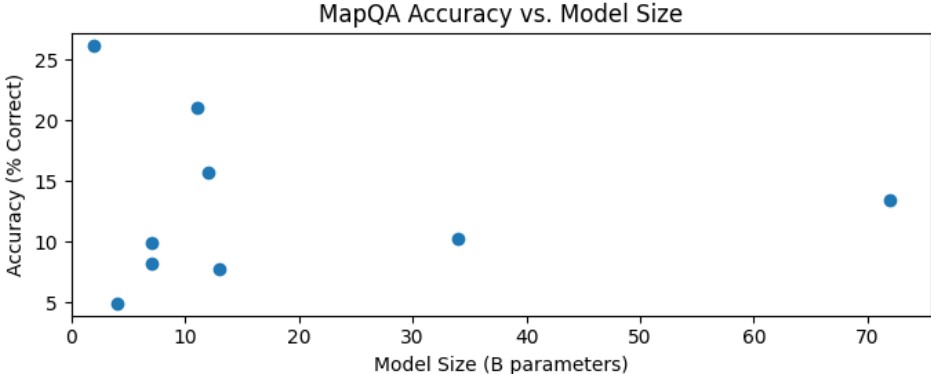

Figure 5: MapQA Benchmark performance compared to model size. Averages are computed per model with each dot representing one model.

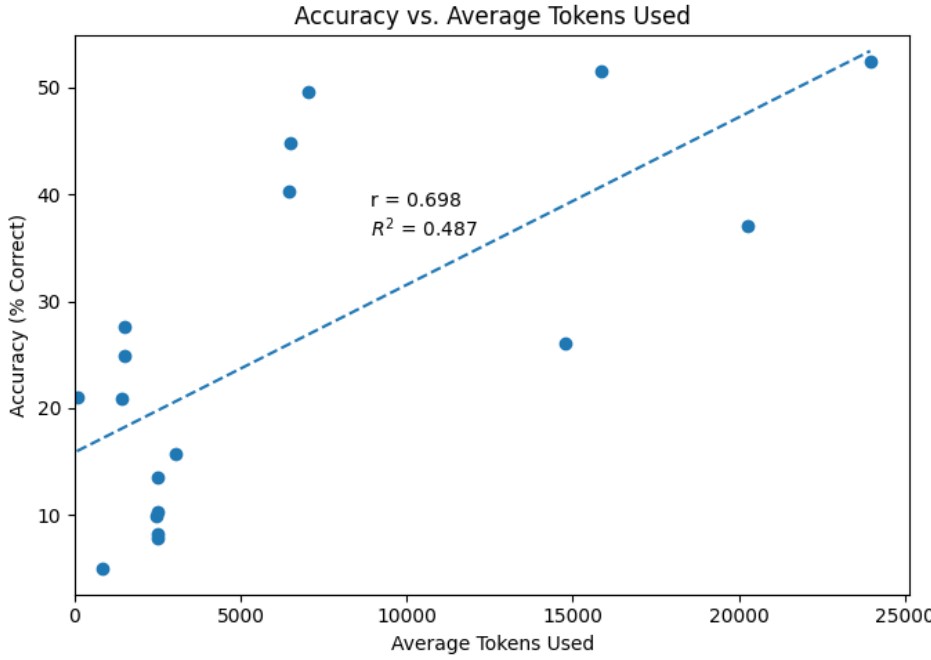

Figure 6: MapQA Benchmark performance compared to tokens used. Input tokens, chain-of-thought tokens, and output tokens are included in the total. Averages are computed per model with each dot representing one model's average accuracy and average number of tokens used. $r = 0.6976$, $p - value = 0.00129$, $R^2 = 0.4867$

Average tokens used are the aggregate of input tokens, chain-of-thought (if any), and output tokens. Unfortunately, the number of chain-of-thought tokens is not individually reported by proprietary models, and is instead categorized like an output token, so deeper analysis is unavailable. In general, models that employ larger image encoder vocabularies and/or use of chain-of-thought achieve higher performance against the MapQA benchmark.

## 5.3 GROUPED DIFFICULTY ACCURACY

Finally, we compare the average model accuracy grouped by difficulty based on user response time. As demonstrated by Mayo et al. (2023), it is common for questions that are easy for humans to also be easy for vision-enabled models. As expected, the Very Easy difficulty group shows the highest

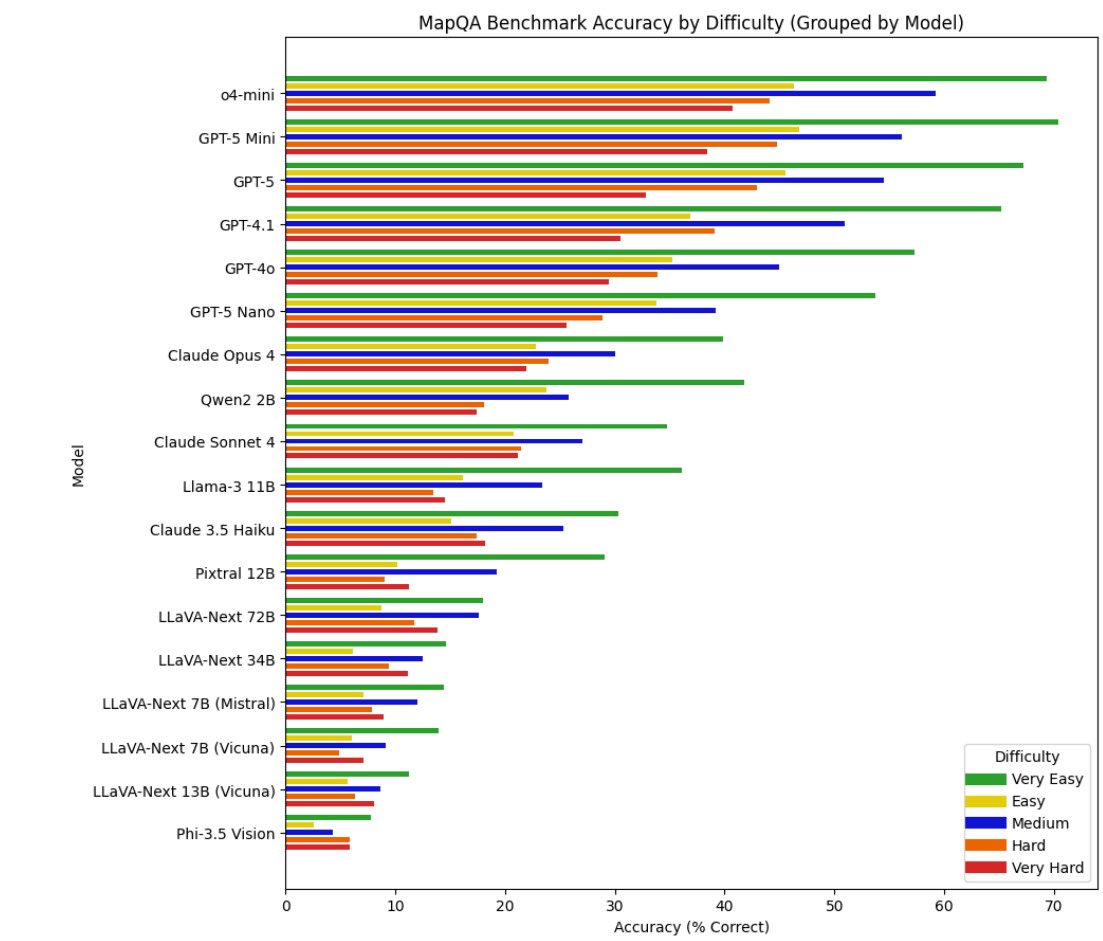

Figure 7: MapQA Benchmark performance for each difficulty.

performance across all models, with GPTo4-mini and GPT5-mini exceeding 70% accuracy. As difficulty (inferred by human response time) increased, the performance of each model decreased with the exception of the Medium difficulty group. More analysis is needed to determine the possible cause of this unexpected outcome.

# 6 CONCLUSION

MapQA establishes the first large-scale, systematically curated benchmark dedicated to evaluating map understanding in large multimodal models. By constructing open-ended, map-dependent questions across diverse domains, MapQA goes beyond text extraction and memorized world knowledge to test genuine spatial reasoning, relational inference, and visual interpretation. Our experiments reveal a striking gap: even the strongest proprietary models fall nearly 40% short of human performance, with open-source models lagging further behind. This persistent shortfall underscores that current LMM progress in charts, diagrams, and general VQA does not yet extend to the unique challenges of cartographic reasoning.

By making these limitations visible, MapQA provides not only a diagnostic tool but also a research agenda. It points to the need for new architectures, training regimes, and evaluation strategies that can integrate symbolic abstraction, spatial reasoning, and visual semantics more effectively. Closing the human–machine gap on map understanding is essential for advancing AI systems capable of supporting high-stakes applications in navigation, environmental science, urban planning, and beyond. MapQA thus serves as a foundation for the next generation of multimodal reasoning research, where genuine map literacy becomes a prerequisite for real-world competence.

## 7 ETHICS

This work complies with the ICLR Code of Ethics. The MapQA benchmark was developed without collecting or storing any personally identifiable information. All map images used are sourced from government archives in the public domain. Human annotators who participated in the curation process were recruited as part of the research team or trained collaborators, and their contributions did not involve sensitive personal data.

Because MapQA focuses on evaluating visual reasoning over maps, potential risks stem from downstream applications of models trained or tested on this benchmark. Misinterpretation of maps by automated systems could have harmful consequences in navigation, disaster response, environmental monitoring, or urban planning. At present, MapQA is released as an evaluation-first benchmark, ensuring that models are rigorously tested before being deployed in such contexts. In the future, we anticipate that MapQA may also serve as a training resource, provided this is done responsibly with attention to ethical safeguards and careful consideration of downstream risks.

## 8 REPRODUCIBILITY

We have taken multiple steps to ensure reproducibility. The MapQA benchmark itself is released in full as part of the supplementary materials (single file), including all image file links, questions, and reference answers. The images are not directly included as they are over 5GB in total, exceeding 100MB limit for OpenReview supplementary material submissions. Images will be directly included in a future github repo along with evaluation scripts upon acceptance. Section 3 (The MapQA Benchmark) provides a detailed description of dataset statistics, splits, and the curation pipeline.

Our evaluation protocol, including the LLM-as-a-Judge setup, calibration procedure, and system prompts, is described in Section 4 (Experiments) and Figure 3. Model inference configurations (devices, sharding strategies, and prompts) are documented in the main text. Baseline accuracies for all tested models are reported in Figures 4, 7, 6, and 7.

To replicate results, researchers need only (i) the released benchmark file, (ii) an evaluation script and images that will be provided in final submission, and (iii) access to the models under test. We note that hardware requirements differ by model size; these details are specified in Section 4.1 (Baselines). Together, these materials enable independent verification of dataset statistics, evaluation results, and analysis.

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

## A    USE OF LLMS

An LLM and other writing tools were used to detect grammar mistakes and typos during the last stages of writing. An LLM was also used to aid in the writing of python script and functional prototyping.

