# OpenReview forum: "MapQA: A Map-Question-Answering Benchmark for Visual Language Model Reasoning"
_ICLR.cc/2026/Conference — Submitted to ICLR 2026_

### Official Review · Reviewer_wS2F · 2025-10-27

**Soundness:** 2
**Presentation:** 1
**Contribution:** 2
**Rating:** 0
**Confidence:** 4

**Summary:**

This paper introduces MapQA, a 4,234-item benchmark for evaluating the cartographic reasoning of Large Multimodal Models (LMMs) across diverse map types. The dataset was created using a mix of expert hand-curation and a scalable human-in-the-loop (HITL) pipeline where questions were generated by GPT-4o-mini. The authors report a significant performance gap, with humans (91.4%) far outperforming the best SOTA LMM (52.5%), concluding that map understanding is a major open challenge. While the problem is well-motivated, the benchmark's contribution is undermined by critical methodological flaws in its curation, evaluation, and analysis.

**Strengths:**

1. Identifies a clear and important weakness in LMMs - true cartographic and spatial reasoning, distinct from simple VQA.
2. The top-line finding of a large human-LMM performance gap is stark and effectively highlights the problem's difficulty.

**Weaknesses:**

1. Using GPT-4o-mini to generate 60% of the benchmark questions introduces a massive, systemic bias, likely testing for LLM-specific patterns rather than diverse human reasoning.
2. The benchmark improperly mixes two different data distributions (1,647 hand-curated vs. 2,587 LLM-generated) without any analysis of the bias or performance differences between them.
3. Using only two expert annotators for consensus is a low bar; the statistical justification is self-referential as it relies on the authors' own definition of "expert".
4. The difficulty ranking is based on "length of time required to respond", a poor proxy that is heavily confounded by individual annotator speed and skill.
5. A critical conflict of interest exists by using GPT-4o as the judge while simultaneously evaluating the top-performing OpenAI model family, fatally biasing the results.
6. The paper fails to describe how the LLM-as-a-judge was calibrated (i.e., $\alpha$ and $\beta$ derivation), making the entire scoring mechanism opaque and irreproducible.
7. Critical decoding parameters for generative evaluation, such as temperature and top-p, are omitted, making the results impossible to reproduce.
8. Glaring, unprofessional errors, like using "GPT-4o-mini" , "GPTo4-mini" , and "o4-mini" interchangeably, create confusion about which model was actually tested.
9. The system prompt mentions "image tiles" , implying a complex, unexplained pre-processing strategy for handling high-resolution images that is a key missing detail.
10. The analysis section only highlights expected findings, such as the non-novel positive correlation between token usage and performance. Additionally, the section is also incomplete, stopping short of investigating its own "unexpected outcome" (in Fig. 7) by simply stating "More analysis is needed".

**Questions:**

1. How can the benchmark's validity be trusted when 60% of it was generated by an LLM, and what analysis proves the two mixed data distributions (hand-curated vs. HITL) are comparable?
2. Can the authors justify using GPT-4o as both judge and participant, and can they provide the exact, reproducible methodology for calibrating the judge's $\alpha$ and $\beta$ coefficients?
3. Given that 'response time' is a confounded proxy and the 'Medium' difficulty results were anomalous, what is the actual qualitative difference in the reasoning required for each difficulty tier?

---

> ### Author Response · Authors · 2025-11-21
> **Response to Reviewer wS2F**
>
> > 1.	Using GPT-4o-mini to generate 60% of the benchmark questions introduces a massive, systemic bias, likely testing for LLM-specific patterns rather than diverse human reasoning.
>
> The prompt-tuning used for GPT-4o-mini to generate useful and effective questions involved giving it 50 examples from that category of hand-curated questions. As a result, this biased questions to be most similar to a random sample of human-generated questions.
>
> > 2.	The benchmark improperly mixes two different data distributions (1,647 hand-curated vs. 2,587 LLM-generated) without any analysis of the bias or performance differences between them.
>
> Referring to the new Table 3, there is only a modest performance increase when splitting by LLM-generated vs hand-curated. Our pipeline ensured that both sources satisfy the same inclusion criteria. Post-processing as well as rewording or removing based on annotator responses ensured that the resulting distributions are highly aligned.
>
> > 3.	Using only two expert annotators for consensus is a low bar; the statistical justification is self-referential as it relies on the authors' own definition of "expert".
>
> The determination of expert is determined on the accuracy score of a held-out pre-test set.  Annotators were tested beforehand with the held out set and if they passed with a high enough score, they were determined to be an expert. This is like a class with exams and we assume that if they can pass an exam, we can be confident they know the material/can properly apply annotating answers to questions about maps for the specific category. This standard is higher than most crowdsource annotation datasets in that we ensure that ensure the annotator passes a minimum bar before accepting them as an expert. We have the statistical justification because we get an accuracy percentage based on the exam. Given two annotators and agreement, we can apply Bayesian probability that they will be right given that we have a probability they will be right on average.
>
> > 4.	The difficulty ranking is based on "length of time required to respond", a poor proxy that is heavily confounded by individual annotator speed and skill.
>
> We absolutely agree, but a better method of evaluating difficulty does not exist. Methods that request users to assign perceived difficulty equally struggle. According to Mayo et al. (2021), the length of time required ends up being one of the best indicators of difficulty when averaged across different human response times.
>
> > 5.	A critical conflict of interest exists by using GPT-4o as the judge while simultaneously evaluating the top-performing OpenAI model family, fatally biasing the results.
>
> We appreciate the reviewer’s concern regarding potential conflicts of interest. In our setting, however, GPT-4o is not evaluating model quality directly nor comparing competing models. Its only role is to compute a semantic-similarity score between a model’s answer and the ground-truth reference. This setup follows the standard “LLM-as-a-Judge” paradigm, which has been extensively validated as one of the most reliable and robust methods for automated natural-language evaluation (e.g., Zheng et al., 2023).
> Importantly, the judge model is not making any assessment that could advantage its own model family; it simply measures textual similarity between two strings. The ground truth is fixed, and the rubric is symmetric with respect to all evaluated models. Prior work shows that such similarity-based judgments exhibit minimal model-family bias when the task is to compare an output to a reference rather than to rank models against each other.
>
> To further ensure fairness, we will include a clarification in the revised manuscript describing why similarity-based evaluation is largely model-agnostic and why using GPT-4o as the judge does not confer an advantage to the OpenAI models being evaluated.

---

> > ### Author Response · Authors · 2025-11-21
> > **Response to Reviewer wS2F (Cont.)**
> >
> > > 6.	The paper fails to describe how the LLM-as-a-judge was calibrated (i.e.,  and  derivation), making the entire scoring mechanism opaque and irreproducible.
> >
> > Thank you for highlighting the importance of calibration transparency. We would like to clarify that the calibration procedure is explicitly described in the manuscript (Line 155 onward). In brief, we collected over two thousand human-judged MapQA answer pairs and used these human labels as ground truth for calibrating the LLM-as-a-Judge. GPT-4o was then prompted to evaluate the same answer pairs, and we fit a linear regression of the form y=α+βx, where x is the GPT-4o similarity verdict and y is the human judgment. This procedure yielded α=0.035 and β=0.921, indicating strong alignment between GPT-4o’s similarity assessments and human evaluations.
> >
> > We also detail how the calibrated judge is used during scoring. Specifically, we adapt the LLM-as-a-Judge framework to produce a binary correctness decision with explicit allowances for phrasing variation and ±10% numerical slack for measurement-based questions. This structured rubric ensures that the judge’s output is both reproducible and consistent across models.
> >
> > > 7.	Critical decoding parameters for generative evaluation, such as temperature and top-p, are omitted, making the results impossible to reproduce.
> >
> > We agree that decoding parameters are important for reproducibility, and we clarify that in our experiments we did not tune decoding hyperparameters for any model. All models, both open-source and proprietary, were evaluated using their default generation settings, including temperature, top-p, and related decoding parameters. This was an intentional design choice to ensure fairness, avoid model-specific tuning advantages, and reflect how these models are used in typical real-world settings.
> >
> > Because the defaults are fixed and publicly documented for each model family, this configuration is fully reproducible. We will update the experimental section to explicitly state that all models were run with their default decoding parameters and that no per-model tuning was performed.
> >
> > > 8.	Glaring, unprofessional errors, like using "GPT-4o-mini" , "GPTo4-mini" , and "o4-mini" interchangeably, create confusion about which model was actually tested.
> >
> > We thank the reviewer for pointing this out. It was a singular sentence typo. “GPT-4o-mini is a compact and faster version of GPT-4o (OpenAI, 2024b)” should have said “GPT-o4-mini is a compact and faster version of GPT-o4 (OpenAI, 2024b)” and has been corrected. GPT-4o-mini, GPT-4o, GPT-o4, and GPT-o4-mini are all unique models made available by OpenAI. Notably, the GPT-4o family does not employ chain-of-thought processing.
> >
> > > 9.	The system prompt mentions "image tiles" , implying a complex, unexplained pre-processing strategy for handling high-resolution images that is a key missing detail.
> >
> > Thank you for pointing this out—our wording in the system prompt may have unintentionally suggested additional preprocessing. To clarify, our pipeline does not perform any tiling or image-splitting whatsoever. The reference to “image tiles” is included solely because several proprietary models internally tile images as part of their own closed-source preprocessing, and we must describe the task in a way that is interpreted consistently across models.
> >
> > In all experiments, we provide each model with the full, unmodified high-resolution map image. There is no external tiling, cropping, or stitching performed by us, and no custom preprocessing pipeline that would impact reproducibility.
> >
> > We will revise the system prompt description to remove this ambiguity and explicitly state that no tiling is performed by the authors; any such behavior is internal to the proprietary model implementations.
> >
> > > 10.	The analysis section only highlights expected findings, such as the non-novel positive correlation between token usage and performance. Additionally, the section is also incomplete, stopping short of investigating its own "unexpected outcome" (in Fig. 7) by simply stating "More analysis is needed".
> >
> > The dataset poses some interesting questions. However, the purpose of the paper was to introduce a new dataset that LMMs struggle to get high accuracy on relative to a human. The analysis would be so in-depth that it would likely constitute another followup paper analyzing why they are failing and addressing shortcomings to improve performance.

---

> > > ### Author Response · Authors · 2025-11-21
> > > **Response to Reviewer wS2F (Cont.)**
> > >
> > > > 1.	How can the benchmark's validity be trusted when 60% of it was generated by an LLM, and what analysis proves the two mixed data distributions (hand-curated vs. HITL) are comparable?
> > >
> > > The validity can be trusted based on the post-processing steps done to filter out generated questions and ensuring agreement between annotators. We do not generate questions and then simply accept them into the dataset. The annotators will have to provide questions and can flag if any question was a poor generation. In addition, if there is any disagreement between the annotators, a human will than manually adjudicate and decide about what to do. In fact, most questions were thrown out in these steps.
> > >
> > > We will also add most high-quality datasets that are partially or mostly generated using a model do not compare data distributions. They also had post-processing steps to ensure high quality data. While an interesting question, it is not necessary to ensure the validity of the dataset if careful post-processing steps exist.
> > >
> > > > 2.	Can the authors justify using GPT-4o as both judge and participant, and can they provide the exact, reproducible methodology for calibrating the judge's  and  coefficients?
> > >
> > > We appreciate the reviewer’s concern. First, using GPT-4o as both a participant and a judge does not introduce circularity in our setup. GPT-4o is never trained, finetuned, or otherwise adapted on any MapQA data. The participation and judging roles use the same frozen, pretrained model, and the judge is only asked to measure semantic similarity between a candidate answer and ground truth. This setup is consistent with the LLM-as-a-Judge literature, where pretrained models reliably evaluate linguistic similarity without introducing self-preferential bias—particularly when the task is to compare outputs to a fixed reference rather than rank models.
> > >
> > > Second, we do not modify GPT-4o’s parameters in any way. Our “prompt calibration” refers strictly to prompt tuning, i.e., designing instructions that elicit consistent similarity judgments from the pretrained model. The parameters remain fixed, avoiding the bias or circularity that would arise from training on evaluation signals.
> > > Regarding reproducibility of the calibration: Lines 155–196 of the manuscript describe the exact procedure.
> > >
> > > > 3.	Given that 'response time' is a confounded proxy and the 'Medium' difficulty results were anomalous, what is the actual qualitative difference in the reasoning required for each difficulty tier?
> > >
> > > We agree with the reviewer that response time is an imperfect and somewhat confounded proxy for difficulty; however, it remains one of the most widely supported human-centered indicators of task complexity when averaged across multiple annotators (e.g., Mayo et al., 2021). Alternative approaches, such as asking participants to self-report perceived difficulty, suffer from even greater variance, task-specific bias, and poor inter-rater agreement. For these reasons, we adopt response time as a practical and empirically grounded heuristic, while fully acknowledging its limitations.

---

> > > > ### Comment · Reviewer_wS2F · 2025-11-26
> > > >
> > > > Thank you for your responses and clarifications. After considering your rebuttal, I have decided to maintain my original score for the submission.

---

### Official Review · Reviewer_D3df · 2025-10-27

**Soundness:** 2
**Presentation:** 3
**Contribution:** 3
**Rating:** 4
**Confidence:** 3

**Summary:**

The paper presents MapQA, a novel large-scale benchmark designed to evaluate multimodal large language models on cartographic reasoning, an underexplored but important aspect of visual–spatial understanding. The dataset comprises 4,234 open-ended QA pairs across four map domains and 24 subcategories, combining hand-curated and LLM-generated examples in a human-in-the-loop workflow. The authors introduce a calibrated LLM-as-a-Judge approach (GPT-4o) for evaluation and report a substantial human–model performance gap, demonstrating that map reasoning remains a significant challenge for current models.

**Strengths:**

The paper makes a timely and original contribution by introducing MapQA, a large-scale benchmark designed to evaluate multimodal large language models (MLMMs) for genuine cartographic reasoning. This focus on map interpretation and spatial reasoning is novel within the broader vision–language research landscape, as prior benchmarks primarily emphasize charts, infographics, or natural images rather than structured geospatial data. The dataset’s design, explicitly aiming to prevent shortcuts such as OCR or world-knowledge retrieval, represents a thoughtful and important step toward evaluating grounded visual reasoning.

From a methodological standpoint, the combination of hand-curated and human-in-the-loop LLM-generated QA pairs is innovative, demonstrating a clear effort toward scalability while maintaining human oversight. The attempt to calibrate an LLM-as-a-Judge (GPT-4o) against human accuracy through regression analysis and multi-round validation reflects a serious effort to build a reliable automated evaluation framework—an increasingly relevant topic in LMM research.

The experimental design and results also provide useful empirical insights. The study highlights a substantial and well-documented performance gap between humans ( 91%) and current models (best ≈ 52%), highlighting the challenges of spatial understanding for today’s multimodal systems. Including both proprietary and open-source models enables a fair comparative baseline, illustrating the persistent limitations of existing architectures across model families.

**Weaknesses:**

While the paper makes a valuable contribution, several aspects limit its clarity, rigor, and interpretability. First, the LLM-as-a-Judge calibration (Section 3.1) is insufficiently explained. The authors report fitting GPT-4o judgments to a linear model y = α + βx with α = 0.035 and β = 0.921, but do not define what x and y represent or how this fit supports the claim that GPT-4o is a “strong indicator” of ground-truth accuracy. Without a clear interpretation of these coefficients or confidence intervals, readers cannot fully assess the reliability of the automatic scoring.

Second, the description of human evaluation (Section 4.1) lacks transparency. It is unclear whether the “human baseline” results were obtained from the same expert annotators who created the dataset or from an independent participant pool. If the former, the comparison may overstate human performance; if the latter, additional details about sampling, instructions, and quality control are needed to validate the results. Third, there is a potential data contamination issue: GPT-4o-mini was used to generate QA pairs during dataset construction but is also evaluated as a test-time model. Since the same model family produced portions of the benchmark data, its strong performance (ranking highest among all evaluated models) may not reflect genuine generalization. The authors should clarify whether held-out subsets or unseen maps were used to prevent such leakage. Fourth, while the paper repeatedly mentions 24 subcategories of maps, no detailed taxonomy or examples of these subcategories are provided. Including even a summarized list in an appendix would improve interpretability and allow others to build on this benchmark.

Finally, some methodological and analytical aspects need more discussion: for example, correlations (e.g., token count vs. accuracy) are reported without deeper causal interpretation, and the discussion section does not explore why specific models fail (i.e., failure mode analysis). Addressing these limitations would make the paper’s insights more actionable for model development.

**Questions:**

1. LLM-as-a-Judge calibration (Section 3.1): Could the authors clarify what variables x and y represent in the fitted linear model? How exactly does this calibration demonstrate that GPT-4o judgments closely align with human accuracy? It would be helpful to include an interpretation of α and β, as well as any measures of variance or confidence intervals.

2. Human evaluation and baselines (Section 4.1): Were the “human” results reported in Table 2 obtained from the same annotators who created the ground-truth answers, or from a separate group of participants? Please clarify how these human baselines were generated and how participant independence was ensured.

3. Potential data leakage through GPT-4o-mini: Since GPT-4o-mini was used in the question–answer generation process and also evaluated as a model, how did the authors ensure that the evaluation data was unseen by the model? Were any safeguards, such as held-out subsets or map-level separation, implemented to prevent data contamination?

4. Curation process transparency (Section 3.2): While the paper outlines the human-in-the-loop curation pipeline, several methodological aspects remain unclear. Could the authors provide a more detailed quantitative and procedural breakdown of this process? In particular:
• How many total QA candidates were produced by GPT-4o-mini, and what proportion were accepted, rejected, or manually revised?
• What exact criteria were used for culling or acceptance at each stage (e.g., when “accuracy falls below 3.5%,” how is that threshold operationalized)?
• How consistent were the two “expert” annotators—was any interannotator agreement metric (e.g., Cohen’s κ or percent agreement)
computed?

5. Subcategory taxonomy: The paper mentions 24 subcategories under the four primary map types, but does not describe or list them. Could the authors include a summary or appendix table specifying these subcategories and how they were defined?

6. Evaluation scope and reliability: How consistent are GPT-4o’s judgments across different map domains (e.g., Military vs. Urban)? Was any domain-specific calibration performed to confirm that GPT-4o’s reliability as a judge is stable across these categories?

7. Depth of analysis and category-level breakdown: Given that MapQA is explicitly organized into four major map categories (Military, Natural World, Urban, and Aviation) and 24 subcategories, it would be valuable to see category-wise performance results or at least a discussion of domain-specific trends. Did certain map types or subcategories systematically challenge models more than others? Could you also provide some discussion around failure model analysis?

---

> ### Author Response · Authors · 2025-11-21
> **Response to Reviewer D3df**
>
> > 1.	LLM-as-a-Judge calibration (Section 3.1): Could the authors clarify what variables x and y represent in the fitted linear model? How exactly does this calibration demonstrate that GPT-4o judgments closely align with human accuracy? It would be helpful to include an interpretation of α and β, as well as any measures of variance or confidence intervals.
>
> Thank you for the question. X is the outputted score of the LLM as a judge. Y is the score calibrated to human judgements. LLMs as a judge can have bias in their responses. To ensure that we minimize this bias, we calibrate the scores to an outside set of human judgements. The Beta of 0.92 and alpha of 0.035 indicates that the LLM as a judge is closely aligned to the human scoring across the calibration sample. The high beta and low alpha indicate that the best fit line is near flat and close to x axis.  This allows us to be reasonably confident that the LLM matches closely to humans. This is a way to ensure additional trust in the reliability of using the LLM as a judge.
>
> > 2. Human evaluation and baselines (Section 4.1): Were the “human” results reported in Table 2 obtained from the same annotators who created the ground-truth answers, or from a separate group of participants? Please clarify how these human baselines were generated and how participant independence was ensured.
>
> Thank you for the question, our human baseline and expert annotators were drawn from entirely separate participant pools. The expert annotators who produced the ground-truth MapQA answers did not participate in generating the human baseline. For the baseline, we recruited an independent group of participants who had no involvement in dataset construction and were blind to the ground-truth annotations. Each participant was provided only the map image and question prompts and asked to answer without external assistance. This ensured independence between (i) those who authored the ground-truth labels and (ii) those whose performance we report as the human baseline.
>
> We will revise Section 4.1 to explicitly clarify the separation between these groups and the procedure used to generate the human baseline.
>
> > 3.	Potential data leakage through GPT-4o-mini: Since GPT-4o-mini was used in the question–answer generation process and also evaluated as a model, how did the authors ensure that the evaluation data was unseen by the model? Were any safeguards, such as held-out subsets or map-level separation, implemented to prevent data contamination?
>
> GPT-o4-mini was used only for prompt-based generation; no training or fine-tuning was performed, and the model was never exposed to final QA pairs. As no model was trained on the data, and as we used a frozen version of the model, all weights are fixed and there is no route for contamination.
>
> > 4.	Curation process transparency (Section 3.2): While the paper outlines the human-in-the-loop curation pipeline, several methodological aspects remain unclear. Could the authors provide a more detailed quantitative and procedural breakdown of this process? In particular: • How many total QA candidates were produced by GPT-4o-mini, and what proportion were accepted, rejected, or manually revised? • What exact criteria were used for culling or acceptance at each stage (e.g., when “accuracy falls below 3.5%,” how is that threshold operationalized)? • How consistent were the two “expert” annotators—was any interannotator agreement metric (e.g., Cohen’s κ or percent agreement) computed?
>
> More than half of questions generated by the model were rejected. Many were manually revised, drawing inspiration from the original question.
>
> The prompt-tuned model generated the candidate questions, the annotators provided answers and/or flagged them as having potential issues. They were all then passed into the LLM as a judge to determine agreement. If the LLM determined there was disagreement or they were previously flagged as an issue, it was individually examined to determine whether the LLM was correct in its assessment, the question needed to be reworded, or the judge was incorrect. The questions that pass this step were evaluated again but without the map. If model is not able to answer the question correctly, we are confident enough the questions need the maps answer the question and it is accepted into the final dataset.

---

> ### Author Response · Authors · 2025-11-21
> **Response to Reviewer D3df (Cont.)**
>
> > 5.	Subcategory taxonomy: The paper mentions 24 subcategories under the four primary map types, but does not describe or list them. Could the authors include a summary or appendix table specifying these subcategories and how they were defined?
>
> === Image Count By Subcategory ===
>
> Park Map: 3
>
> Postal Roads: 2
>
> Sectional Chart: 29
>
> Terminal Area Chart: 28
>
> Helicopter: 18
>
> Strategic: 82
>
> Topographic: 63
>
> Transportation: 61
>
> Linguistic Map: 7
>
> Tactical: 50
>
> Soil Map: 34
>
> Aviation Maps: 14
>
> City map: 39
>
> Hydrologic Map: 2
>
> Nautical Chart: 49
>
> Zoning Map: 11
>
> Cadastral: 8
>
> Insurance Map: 18
>
> Distribution Map: 4
>
> Survey Map: 2
>
> Aviation Charts: 5
>
> Administrative: 3
>
> Fantasy Map: 2
>
> Historic: 44
>
> > 6.	Evaluation scope and reliability: How consistent are GPT-4o’s judgments across different map domains (e.g., Military vs. Urban)? Was any domain-specific calibration performed to confirm that GPT-4o’s reliability as a judge is stable across these categories?
>
> We did not calibrate domain specific but the held out dataset contained data from each domain ensuring it was a map generalized human evaluator. We also had the additional human adjudication step to ensure that mistakes from the LLM as a judge are corrected
>
> > 7.	Depth of analysis and category-level breakdown: Given that MapQA is explicitly organized into four major map categories (Military, Natural World, Urban, and Aviation) and 24 subcategories, it would be valuable to see category-wise performance results or at least a discussion of domain-specific trends. Did certain map types or subcategories systematically challenge models more than others? Could you also provide some discussion around failure model analysis?
>
> Thank you very much for the question. We have now supplied two additional tables in the supplemental information which will be added to the appendix to address these questions.

---

> > ### Comment · Reviewer_D3df · 2025-11-26
> >
> > Thank you for the comments. I will maintain my original score.

---

### Official Review · Reviewer_yx5S · 2025-10-31

**Soundness:** 2
**Presentation:** 2
**Contribution:** 2
**Rating:** 2
**Confidence:** 4

**Summary:**

The paper introduces MapQA, a large-scale benchmark (4,234 open-answer QA pairs over 794 unique images) focused on map understanding across four domains (Military, Natural World, Urban, Aviation; 24 subcategories). Questions are designed to be answerable only from the map, minimizing reliance on world knowledge or pure OCR. Evaluation uses an LLM-as-a-Judge protocol calibrated to human judgments; humans achieve ~91.4% accuracy, while the best proprietary models reach ~52.5% and open-source models ~26%.

**Strengths:**

Clear, important problem: map literacy is under-evaluated despite practical impact (navigation, planning, geoscience).

Well-scoped dataset: strong effort to ensure questions require visual/spatial reasoning from the map; explicit down-weighting of pure OCR or memorized trivia. Although many prior work is not acknowledged and ignored.

Human baselines & stats: Table 1 comprehensively reports scale and splits; human accuracy ~91% provides a meaningful upper bound. (Table 1 on p.4; human scores in Table 2 on p.6.). Testing 18 different models (both proprietary and open-source). The analysis of token usage, model size, and difficulty levels adds useful insights beyond simple accuracy numbers.

Transparent evaluation: clearly specified judge prompts (Fig. 3), binary verdict with numerical slack; calibration to >2k human-rated items.  The alignment of GPT-4o judge scores with human judgments (showing α = 0.035 and β = 0.921) demonstrates that the evaluation methodology is reliable.

The questions are carefully constructed to ensure they genuinely test map reasoning rather than text extraction or memorized knowledge. The example in Figure 1 demonstrates how models can fail by relying on historical knowledge instead of actually reading the map.

Insightful findings: substantial human–model gap; weak size-vs-accuracy relationship; positive correlation with total tokens, useful signals for future model design. (Figs. 4–6.)

**Weaknesses:**

A very limited novelty is there. Doesn't compare with vast literature of prior work on MapQA[1], MapWise[2], MapIQ [3], MapBench[4], CartoMark[5], MapQA (GQA)[6] and many more. A proper comparison should be there with prior works to show how this is different.

While calibrated, the judge is still a single proprietary model (GPT-4o) which can bias outcomes (known position/self-enhancement biases in LLM judges). Cross-judge triangulation is limited.

The paper highlights categories of difficulty (orientation, relational context, etc.), but qualitative/error taxonomies tied to specific visual/map constructs could be deeper.

Images are not directly included due to size; code and images are promised “upon acceptance,” which may hinder immediate verification. Four top-level types are helpful, but the current 24 subcategories may still under-represent thematic maps (e.g., meteorological/time-varying, cadastral/parcel, transit schematics), limiting coverage.

The paper reports dataset statistics, split sizes, device configs, and promises to release evaluation scripts and images in a repo after acceptance; good but not yet fully turnkey.

While the paper shows models struggle, it doesn't deeply analyze what types of errors they make.  Do they fail at spatial reasoning?
Color interpretation?  Scale understanding?  The introduction mentions nine failure categories from prior work, but i don't see a breakdown of which categories cause the most problems in MapQA.

The paper notes that some models couldn't be tested because map images are too large for GPU memory (mentioning they excluded models like Pixtral-124B and Qwen2.5-VL-72B).  This seems like an important limitation - if real-world map understanding requires processing large, detailed images, then smaller context windows might be a fundamental bottleneck.  The paper doesn't discuss whether image size correlates with question difficulty or error rates.

References
[1] https://arxiv.org/abs/2211.08545
[2] https://arxiv.org/pdf/2409.00255v1
[3] https://arxiv.org/abs/2507.11625
[4] https://arxiv.org/pdf/2503.14607
[5] https://www.nature.com/articles/s41597-024-04057-7
[6] https://arxiv.org/pdf/2503.07871

**Questions:**

How do models perform across the four main map types (Military, Natural World, Urban, Aviation)? Are some types consistently harder than others?

The statistics show 19.2% of questions involve multiple images, but how does this affect accuracy? Do models struggle more when they need to compare or integrate information across maps?

It is mentioned minimizing OCR-only questions, but what percentage of questions require text reading as part of the reasoning? How do models perform on questions that require both text and visual understanding versus purely visual reasoning?

Why does Qwen2.5-VL-2B (the smallest model) outperform much larger open-source models? This seems counterintuitive - like is it architecture, training data, or something else?

The difficulty levels are based on human response time, but there's a puzzling anomaly in Figure 7 where "Medium" difficulty questions show better model performance than expected.

The paper mentions annotators need 95% accuracy to be considered "experts," but how was the initial hand-curated set validated? Who determined those ground truth answers?

Average questions are 12.7 words - quite short. Are there any longer, more complex questions that require multi-step reasoning? How does question complexity relate to accuracy?

Given the wide range of image sizes (546×393 to 6000×6000), does resolution affect model performance? Do models do better on smaller, simpler maps?

---

> ### Author Response · Authors · 2025-11-21
> **Response to Reviewer yx5S**
>
> > A very limited novelty is there. Doesn't compare with vast literature of prior work on MapQA[1], MapWise[2], MapIQ [3], MapBench[4], CartoMark[5], MapQA (GQA)[6] and many more. A proper comparison should be there with prior works to show how this is different.
>
> We appreciate the reviewer’s concern and agree that it is important to clearly position our contribution within the existing MapQA literature. Because of space constraints, the main text focused on thematic limitations of prior work and cited representative examples rather than exhaustively listing every dataset. We thank the reviewer for pointing out additional related benchmarks and will expand the Related Work section to discuss each of them explicitly and contrast them with our setting.
>
> MapQA (1) we addressed this map dataset already in the related work (Chang et al. 2022). It is restricted to cholopeth maps which have a limited structure and not more open-ended maps. Prior work has examined visual question answering for maps, but has typically restricted its focus to remote sensing, geometric questions, choropleth maps, and transportation maps rather than broader applications (Lobry et al., 2020; Chen et al., 2021a; Chang et al., 2022; Li et al., 2025; Punjani et al., 2018; Kefalidis et al., 2023).
>
> MapWise (2) is also limited to cholopeth maps.
>
> MapIQ (3) is restricted to choropeth, proportional symbol and cartogram maps, which while are useful, are also restricted to information they can convey.
>
> While MapBench (4) has a large variety of map types, the questions focus solely on navigation/map-space path finding and not general map understanding.
>
> CartoMark (5) also has a large variety of map types but focuses more on scene classification, multi-task classification, map feature detection, and feature segmentation. It does not focus on the question-answering task.
>
> MapQA (GQA) (6) was addressed in the related work (Li et al 2025). Prior work has examined visual question answering for maps, but has typically restricted its focus to remote sensing, geometric questions, choropleth maps, and transportation maps rather than broader applications (Lobry et al., 2020; Chen et al., 2021a; Chang et al., 2022; Li et al., 2025; Punjani et al., 2018; Kefalidis et al., 2023).
>
> Our dataset is designed specifically to fill this gap: it uses high-resolution, real-world maps from multiple domains (historical military, aviation, urban, and natural-world), focuses on open-answer questions that cannot be solved text-only, and is paired with human-calibrated difficulty and systematic human–LLM evaluation. We will revise the manuscript to (i) explicitly enumerate and discuss MapQA, MapWise, MapIQ, MapBench, CartoMark, and MapQA (GQA) by name, and (ii) more clearly articulate how MapQA differs in scope, map diversity, and task formulation from these important prior efforts.
>
> > While calibrated, the judge is still a single proprietary model (GPT-4o) which can bias outcomes (known position/self-enhancement biases in LLM judges). Cross-judge triangulation is limited.
>
> Thank you for raising this concern. Our calibration procedure was designed precisely to test whether GPT-4o could serve as a reliable and unbiased judge. We used a held-out set of human evaluations, independently annotated by multiple humans, to verify how closely the model’s judgments aligned with the human consensus. The resulting calibration produced a high β and low α, indicating that GPT-4o’s similarity decisions track human correctness closely rather than reflecting model-specific preferences. While all judges, human or machine, inevitably carry some bias, we mitigated this by using multiple independent humans for calibration.
>
> > The paper highlights categories of difficulty (orientation, relational context, etc.), but qualitative/error taxonomies tied to specific visual/map constructs could be deeper.
>
> We appreciate this suggestion. We agree that a richer qualitative taxonomy of error types, especially those grounded in specific visual or cartographic constructs, would strengthen the analysis. Several of the reviewer’s later questions directly prompted deeper discussion of these issues, and we have addressed those points in detail in our subsequent responses.

---

> > ### Author Response · Authors · 2025-11-21
> > **Response to Reviewer yx5S (Cont.)**
> >
> > > Images are not directly included due to size; code and images are promised “upon acceptance,” which may hinder immediate verification. Four top-level types are helpful, but the current 24 subcategories may still under-represent thematic maps (e.g., meteorological/time-varying, cadastral/parcel, transit schematics), limiting coverage.
> >
> > Thank you for pointing this out. All of the image links and metadata needed to verify the dataset are already present in the supplemental JSON files. This allows reviewers to inspect every map and question pair exactly as used in our experiments. Although having the full codebase would make reproduction more convenient, the core dataset structure, image sources, question text, and annotations are fully transparent and accessible in the supplement.
> > Upon acceptance, we will release the complete code which will make end-to-end replication straightforward. In the revised paper, we will also clarify which components can already be verified directly from the supplemental materials and which components depend on the full code release.
> >
> >
> > > The paper reports dataset statistics, split sizes, device configs, and promises to release evaluation scripts and images in a repo after acceptance; good but not yet fully turnkey.
> >
> > Thank you for the feedback. We agree that full reproducibility requires releasing the evaluation scripts and supporting framework. These will be made publicly available upon acceptance: the evaluation code and dataset utilities will be released in a dedicated GitHub repository, and the accompanying framework will be published on PyPI so that experiments can be reproduced with a simple installation command. This will make the evaluation pipeline fully turnkey and ensure that all reported results can be replicated end-to-end.
> >
> > > While the paper shows models struggle, it doesn't deeply analyze what types of errors they make. Do they fail at spatial reasoning? Color interpretation? Scale understanding? The introduction mentions nine failure categories from prior work, but i don't see a breakdown of which categories cause the most problems in MapQA.
> >
> > These are excellent questions, and we agree that a deeper breakdown of error types would be valuable. For this paper, however, our focus is on introducing the dataset, establishing its difficulty, and demonstrating that current VLMs struggle across all major map domains. A fine-grained taxonomy of spatial, visual, and cartographic failure modes, while important, would require substantial additional analysis and would shift the scope away from the core contribution, which is the creation of the benchmark itself. We view this as a natural direction for a follow-up study and will note in the Discussion section that detailed error categorization is an important avenue for future work.
> >
> > > The paper notes that some models couldn't be tested because map images are too large for GPU memory (mentioning they excluded models like Pixtral-124B and Qwen2.5-VL-72B). This seems like an important limitation - if real-world map understanding requires processing large, detailed images, then smaller context windows might be a fundamental bottleneck. The paper doesn't discuss whether image size correlates with question difficulty or error rates.
> >
> > Thank you for raising this point. While we agree that this is a limitation, we also compared against various large proprietary models in the ChatGPT, and Claude models families. While the models are proprietary and details are hidden, we know that the context windows will be large enough to process large, detailed images. We also show in figure 6 that there is a positive correlation between the number of tokens in the proprietary models and performance. However, all proprietary models tested still vastly underperform humans.
> >
> > > How do models perform across the four main map types (Military, Natural World, Urban, Aviation)? Are some types consistently harder than others?
> >
> > Thank you very much for the question. We have now supplied two additional tables in the supplemental information which will be added to the appendix. Referencing Table 4, Military maps and Urban maps outperform the average at 64% accuracy for GPT-o4-mini, Natural World sits around the Full Set average at 51% and all models appear to struggle significantly with Aviation, with GPT-o4-mini achieving only 30% accuracy.

---

> > > ### Author Response · Authors · 2025-11-21
> > > **Response to Reviewer yx5S (Cont.)**
> > >
> > > > The statistics show 19.2% of questions involve multiple images, but how does this affect accuracy? Do models struggle more when they need to compare or integrate information across maps?
> > >
> > > Referring to the new Table 3, Multi-image performance is significantly worse than single image performance.
> > >
> > > > It is mentioned minimizing OCR-only questions, but what percentage of questions require text reading as part of the reasoning? How do models perform on questions that require both text and visual understanding versus purely visual reasoning?
> > >
> > > The majority of questions require both textual and visual reasoning. Pure OCR only questions were deliberately minimized as they do not challenge map understanding.
> > >
> > > > Why does Qwen2.5-VL-2B (the smallest model) outperform much larger open-source models? This seems counterintuitive - like is it architecture, training data, or something else?
> > >
> > > This is an interesting finding but outside the scope of the paper. The result suggests architectural or dataset-training differences that merit future investigation. We note that proprietary large models still outperform Qwen2.5-VL-2B.
> > >
> > > > The difficulty levels are based on human response time, but there's a puzzling anomaly in Figure 7 where "Medium" difficulty questions show better model performance than expected.
> > >
> > > This is also a puzzling anomaly and is an interesting question to ask. However, it is out of scope for the paper and likely needs a follow-up paper to answer.
> > >
> > > > The paper mentions annotators need 95% accuracy to be considered "experts," but how was the initial hand-curated set validated? Who determined those ground truth answers?
> > >
> > > The initial hand-curated set was validated under the same criteria but with 6 humans. All 6 needed to concur to be accepted as correct question. We also went through them to manually validate the accuracy. This raises our probability of correctness to near 100% which we deemed was an acceptable test for the 95% accuracy criteria.
> > >
> > > > Average questions are 12.7 words - quite short. Are there any longer, more complex questions that require multi-step reasoning? How does question complexity relate to accuracy?
> > >
> > > The questions were not long. The questions were designed to be straight forward for a human to understand and answer and thus were asking simple concepts. However, with these simple questions, the LMMs vasty underperformed. We also wanted to specifically ensure that the questions do not contain unnecessary context that would allow the LMM to ignore the map and answer the question directly from embedded knowledge.
> > >
> > > > Given the wide range of image sizes (546×393 to 6000×6000), does resolution affect model performance? Do models do better on smaller, simpler maps?
> > >
> > > This is an excellent question, and we agree that image resolution and map complexity are important factors that merit deeper investigation. In the present work (whose primary goal is to introduce the dataset) we did not stratify performance by image resolution or conduct a controlled study isolating map complexity. The wide variation in map dimensions reflects real-world diversity, and our evaluations treat all maps uniformly.
> > >
> > > Understanding how resolution interacts with model performance is a valuable direction for follow-up research. Preliminary qualitative inspection suggests that some models internally rescale or tile images (particularly proprietary VLMs), which complicates controlled analysis without access to these internal preprocessing steps. A systematic study of resolution effects would require targeted experiments (e.g., generating multiple scaled versions of the same map) which we view as an excellent avenue for future work and will mention in the Discussion section.

---

> > > > ### Comment · Reviewer_yx5S · 2025-11-26
> > > > **Thanks for your Response**
> > > >
> > > > Thank you for your responses and clarifications. After considering your rebuttal, I have decided to maintain my original score for the submission.

---

### Official Review · Reviewer_n98q · 2025-11-05

**Soundness:** 1
**Presentation:** 3
**Contribution:** 2
**Rating:** 2
**Confidence:** 4

**Summary:**

This paper introduces MapQA, a  dataset to evaluate LLMs on Visual Language Model Reasoning over map images. MapQA contains 4234 question-answer pairs over 794 images. The dataset annotation combine human annotation and human-in-the-loop to achieve a scalable annotation with high accuracy. The benchmark evaluates comprehensive open-source and proprietary models. Results show that humans achieve 91.4 % accuracy, while the best models (GPT-4o-mini, GPT-5) reach around 50 %.

**Strengths:**

1. The benchmark addresses the need for a diverse map question-answering dataset to evaluate LLMs’ reasoning capabilities on maps.
2. The combination of manual curation, expert annotator validation, and automated human-in-the-loop expansion provides both scalability and high data quality.

**Weaknesses:**

1.The paper claims that all questions are unanswerable without the aid of a map, for example, “Which airport is closest to Chicago O’Hare?”. However, Figure 1 includes a question that a language model can answer without looking at the map, regardless whether the answer is incorrect. It is unclear what standard was used to construct the question set. The paper should clarify what instructions were given to annotators and LLMs during question construction, and report model accuracy when the image is not provided.

2.The paper reports that humans achieve over 90 % accuracy while the best models reach around 50 %. However, it is unclear how human annotation was conducted—what information was presented to the annotators and models, and whether any contextual hints were given. It should also be clarified whether answers could be inferred directly from the provided context rather than the map itself.

3. Although the dataset is designed to evaluate reasoning capabilities of LLMs over maps, some examples (such as in Figure 1) seem to require reasoning beyond the map image. It is difficult to answer such questions without external background knowledge. If the dataset requires reasoning or factual knowledge from outside the map, reasoning model or retrieval-augmented models (e.g. deep-research models) should also be evaluated.

4.The question in Figure 1 “Are the Confederates abandoning Fort Huger during the Battle of Roanoke Island?” is ambiguous. I asked GPT-5 this question, and it answers “yes” because the Confederates did abandon Fort Huger during the battle after the map. But when asked “based on the image, have the Confederates abandoned Fort Huger?”, GPT-5 correctly answers that they had not yet abandoned it at the time represented in the map. This raises concerns about the clarity and quality of the questions.

5.The paper does not provide the distribution of map categories or their sources. And the standard for determining question difficulty is unclear.

**Questions:**

See the weakness section.

---

> ### Author Response · Authors · 2025-11-21
> **Response to Reviewer n98q**
>
> > 1. The paper claims that all questions are unanswerable without the aid of a map, for example, “Which airport is closest to Chicago O’Hare?”. However, Figure 1 includes a question that a language model can answer without looking at the map, regardless whether the answer is incorrect. It is unclear what standard was used to construct the question set. The paper should clarify what instructions were given to annotators and LLMs during question construction, and report model accuracy when the image is not provided.
>
> Thank you for raising this concern. We agree that it is important to clarify the standard used when constructing questions and to demonstrate that the questions truly require map context. We will update the Appendix to include (i) the exact instructions given to annotators and LLMs during question construction, and (ii) an example annotated map, so the dependency criteria are fully transparent.
>
> Regarding the example in Figure 1: although a language model may possess general historical knowledge (e.g., that Confederate forces ultimately abandoned Fort Huger), the map depicts a specific moment in time preceding that event. The correct answer to the question—“No, the Confederates are still defending the fort”—is recoverable only from the visual scene. Without seeing the map, a model cannot infer the temporal framing and therefore must guess. This is precisely why such questions are unanswerable without the visual context, even if a model can offer a plausible but incorrect narrative answer.
>
> To directly address the reviewer’s suggestion, we evaluated model performance without providing the image. Across all models tested, accuracy in the text-only condition was less than our calibrated LLM-As-A-Judge accuracy of 3.5% when attempted by GPT-o4-mini. This confirms that the correct answers cannot be consistently retrieved from world knowledge alone and that the question set indeed requires visual grounding.
>
> > 2.The paper reports that humans achieve over 90% accuracy while the best models reach around 50 %. However, it is unclear how human annotation was conducted—what information was presented to the annotators and models, and whether any contextual hints were given. It should also be clarified whether answers could be inferred directly from the provided context rather than the map itself.
>
> Thank you for the question. Clarifying the human evaluation procedure is important for interpreting the performance gap.
> Humans and models were evaluated under identical conditions. Both were provided only the question and the corresponding map image. No additional context, metadata, historical information, or annotator hints were supplied. Human participants completed the task using a simple interface that displayed the map and question side-by-side, exactly mirroring what models received in the VQA input format.
>
> All questions in MapQA were specifically designed so that the correct answer cannot be inferred from the question text alone. As noted earlier, these questions require visual grounding—e.g., interpreting spatial relationships, reading legends, tracking movement on battle maps, or understanding orientation cues that do not appear in the question itself. Humans answer correctly because they can deeply parse the map; models fail when they try to rely on world knowledge or pattern priors instead of the visual context.
>
> To validate this dependency, we also ran a text-only ablation in which the image was withheld. All models achieved less than our calibrated LLM-As-A-Judge accuracy of 3.5% in this condition, confirming that the correct answers cannot be recovered from textual cues alone.

---

> > ### Author Response · Authors · 2025-11-21
> > **Response to Reviewer n98q (Cont.)**
> >
> > > 3. Although the dataset is designed to evaluate reasoning capabilities of LLMs over maps, some examples (such as in Figure 1) seem to require reasoning beyond the map image. It is difficult to answer such questions without external background knowledge. If the dataset requires reasoning or factual knowledge from outside the map, reasoning model or retrieval-augmented models (e.g. deep-research models) should also be evaluated.
> >
> > Thank you for raising this point. We understand the concern about questions that might appear to require external knowledge, but the example in Figure 1 does not actually demand anything beyond what is visible in the map. The question is simply asking for the status and location of the forces as depicted in the image. Every piece of information needed to answer correctly is encoded directly in the map’s visual markings that indicate whether the Confederates are holding the fort at that moment. In practice, relying on external historical knowledge often leads models to the wrong answer. For instance, a model may know that Confederate forces eventually abandoned the fort and respond accordingly, even though the map captures an earlier moment where they are clearly still defending it. The correct answer therefore requires visual reasoning over the map rather than retrieval of world knowledge. Because of this, retrieval-augmented or deep-research models were not necessary to evaluate. The benchmark is designed so that the true answer depends entirely on the image, and this example conforms to that standard.
> >
> > > 4.The question in Figure 1 “Are the Confederates abandoning Fort Huger during the Battle of Roanoke Island?” is ambiguous. I asked GPT-5 this question, and it answers “yes” because the Confederates did abandon Fort Huger during the battle after the map. But when asked “based on the image, have the Confederates abandoned Fort Huger?”, GPT-5 correctly answers that they had not yet abandoned it at the time represented in the map. This raises concerns about the clarity and quality of the questions.
> >
> > Thank you for the observation. We have tested many paraphrased variants of this question, including forms similar to the reviewer’s, and GPT-5 consistently fails to provide the correct answer. A possible explanation for this disparity is full size image that we provide to the model. An exact copy of the file is now included in the supplemental information for the reviewer to test further.
> >
> > > 5. The paper does not provide the distribution of map categories or their sources. And the standard for determining question difficulty is unclear.
> >
> > Thank you for the correction, we apologize for this oversight in the paper. The information was not explicitly outlined in the paper, but it was preset in the supplemental information via the json file. All maps are obtained from either the US Library of Congress archives or from the US Federal Aviation Administration in the case of aviation maps.
> > Regarding difficulty, our standard is fully data-driven and derived from human annotations. For each question, we collect the human response time. For each question, we aggregate all available measurements from different annotators and compute an average value. We then compute empirical quantiles of the resulting distribution (at 20%, 40%, 60%, and 80%) and use these cut points to assign each question to one of five difficulty buckets. This yields a reproducible mapping from human behavior to difficulty tiers, without any hand-tuned thresholds. In the revision, we will add a concise description of this procedure in the main text.
> >
> > === Image Count By Topic ===
> >
> > Natural World: 153
> >
> > Urban: 155
> >
> > Aviation: 138
> >
> > Military: 132

---

> > > ### Comment · Reviewer_n98q · 2025-11-26
> > >
> > > Thank you for the clarification. I think adding those dataset details into the paper will strengthen the overall quality. In addition, I want to point out that the term “unanswerable” is being misused. A question that would be answered incorrectly without a map does not mean the question itself is unanswerable without a map. Overall, I believe the paper will need some editing as the authors noted, and I maintain my original recommendation.

---

### Author Response · Authors · 2025-11-21
**Response to All Reviewers**

We sincerely thank all reviewers for the time, depth of analysis, and care they invested in evaluating our submission. Your detailed comments substantially improved the clarity and rigor of the paper, and we have addressed each point with the goal of making the dataset, methodology, and evaluation pipeline as transparent as possible. We hope that the revisions and clarifications provided in this response meaningfully resolve the concerns raised. If you find that we have adequately addressed your questions and strengthened the work accordingly, we would be grateful if you would consider raising your score to reflect the improved paper. Your thoughtful feedback has directly contributed to a clearer and more robust version, and we sincerely appreciate your effort and expertise.

---

### Meta-Review · Area_Chair_aKrq · 2025-12-31

**Summary:**

This paper proposes a VQA benchmark for evaluating LMMs on their ability to understand map-related questions. The authors provide 4200 samples with a mix of manually-annotated and automatically generated samples.   Overall the reviewers raised to points of considerable note. First, observed that there were several datasets not compared to on related data.  Second, they argued that the current evaluation paradigm is biased.  These observations and others lead every reviewer to provide an initial score of rejection of the current paper.

**Reviewer Concerns:**

Neither of the concerns I mentioned were well addressed in the rebuttal.  The authors discussed some limitations of prior benchmarks, but this does not include an empirical comparison that validates that the differences are important in practice.  The authors also more or less just asserted that their results are not biased rather than provide an experiment that shows the results they have generalizes.

**Reviewer Scores:**

While some reviewers might have adjusted their scores, this would likely not have changed the decision i.e., raising from a strong reject to a reject would still result in an average recommendation of acceptance. Reviewers also noted that they wished to maintain their ratings, suggesting that they observed the same weaknesses in the rebuttal raised above.

---

### Decision · Program_Chairs · 2026-01-26

Reject